# Structured and Deep Similarity Matching via Structured and Deep Hebbian Networks

**Dina Obeid**      **Hugo Ramambason**      **Cengiz Pehlevan**

John A. Paulson School of Engineering and Applied Sciences
Harvard University
Cambridge, MA, USA

`{dinaobeid@seas,hugo_ramambason@g,cpehlevan@seas}.harvard.edu`

## Abstract

Synaptic plasticity is widely accepted to be the mechanism behind learning in the brain's neural networks. A central question is how synapses, with access to only local information about the network, can still organize collectively and perform circuit-wide learning in an efficient manner. In single-layered and all-to-all connected neural networks, local plasticity has been shown to implement gradient-based learning on a class of cost functions that contain a term that aligns the similarity of outputs to the similarity of inputs. Whether such cost functions exist for networks with other architectures is not known. In this paper, we introduce structured and deep similarity matching cost functions, and show how they can be optimized in a gradient-based manner by neural networks with local learning rules. These networks extend Földiak's Hebbian/Anti-Hebbian network to deep architectures and structured feedforward, lateral and feedback connections. Credit assignment problem is solved elegantly by a factorization of the dual learning objective to synapse specific local objectives. Simulations show that our networks learn meaningful features.

## 1 Introduction

End-to-end training of neural networks by gradient-based minimization of a cost function has proved to be a very powerful idea, prompting the question whether the brain is implementing the same strategy. The problem with this proposal is that synapses, the sites of learning in the brain, have access to only local information, i.e. states of the pre- and post-synaptic neurons, and neuromodulators, which may represent, for example, global error signals [1]. How can synapses calculate from such incomplete information the necessary partial derivatives, which depend on non-local information about other neurons and synapses in the network? Researchers have been tackling this problem by searching for a biologically-plausible implementation of the backpropagation algorithm [2]. While significant progress has been made in this domain, see e.g. [3, 4, 5, 6, 7, 8, 9, 10, 11, 12], a fully plausible implementation is not yet available.

Here we take another approach and focus on networks already operating with biologically-plausible local learning rules. We ask whether one can formulate network-wide learning cost functions for such networks and whether these networks achieve efficient "credit assignment" by performing gradient-based learning. Previous work in this area showed that single-layered, all-to-all connected Hebbian/anti-Hebbian networks minimize various versions of similarity matching cost functions [13, 14, 15]. In this paper, we generalize these results to networks with structured connectivity and deep architectures.

To achieve our goal, we first introduce a novel class of unsupervised learning objectives that generalize similarity matching [16, 13]: structured and deep similarity matching. This generalization allows for making use of spatial structure in data and hierarchical feature extraction. A parallel can be drawn to the extension of sparse coding [17] to structured [18] and deep sparse coding [19, 20, 21].

We show that structured and deep similarity matching can be implemented by a new class of multi-layered neural networks with structured connectivity and biologically-plausible local learning rules. These networks have Hebbian learning in feedforward and feedback connections between different layers, and anti-Hebbian learning in lateral connections within a layer. They generalize Földiak's single-layered, all-to-all connected Hebbian/anti-Hebbian network [22].

We show how efficient credit assignment is achieved in structured and deep Hebbian/anti-Hebbian networks in an elegant way. The network optimizes a dual min-max problem to the structured and deep similarity matching problem. The network-wide dual objective can be factorized into a summation of distributed objectives over each synapse that depend only on local variables to that synapse. Therefore, gradient learning on them leads to local Hebbian and anti-Hebbian learning rules. Previous work showed this result in a single-layered, all-to-all connected Hebbian/anti-Hebbian network [14]. Here, we extend the result to multi-layered architectures with structured connectivity.

The rest of this paper is organized as follows. In Section 2, we review and extend the results on similarity matching cost functions and their relation to single-layered, all-to-all connected Hebbian/anti-Hebbian networks. In Section 3, we introduce structured similarity matching and in Section 4 we extend it to deep architectures. In Section 5, we derive structured and deep Hebbian/anti-Hebbian networks from these cost functions, and show how credit assignment is achieved. We show simulation results in Section 6 and conclude in Section 7.

## 2 Similarity matching and gradient-based learning in a Hebbian/anti-Hebbian network

In this section we review the results of [14] on Hebbian/anti-Hebbian neural networks and extend them to general monotonic activation functions. Földiak introduced the Hebbian/anti-Hebbian network as a biologically-plausible, single-layered, competitive unsupervised learning network that forms sparse representations [22]. Given an input $\mathbf{x} \in \mathbb{R}^K$, first, the network produces an output, $\mathbf{r} \in \mathbb{R}^N$, by running the following recurrent dynamics until convergence to a fixed point,

$$\tau \frac{d\mathbf{u}(s)}{ds} = -\mathbf{u}(s) + \mathbf{W}\mathbf{x} - (\mathbf{L} - \mathbf{I})\,\mathbf{r}(s),$$
$$\mathbf{r}(s) = \mathbf{f}(\mathbf{u}(s)), \tag{1}$$

where $f$ is the activation function and $\tau$ is the time constant of the neural dynamics. Given the fixed point output, $\mathbf{r}^*$, the synaptic weights are updated by the following local learning rules,

$$\Delta W_{ij} = \eta\left(r_i^* x_j - W_{ij}\right), \quad \Delta L_{ij} = \frac{\eta}{2}\left(r_i^* r_j^* - L_{ij}\right), \tag{2}$$

where $\eta$ is the learning rate. $W_{ij}$ updates are Hebbian synaptic plasticity rules with a linear decay term. $L_{ij}$ updates are anti-Hebbian, because of the minus sign in the corresponding term in (1), and implement lateral competition. After the synaptic update, the network takes in the next input, and the whole process is repeated.

When activation functions are linear, rectified-linear or shrinkage functions, previous studies [23, 13, 14] showed that the learning rules of this network can be interpreted as a stochastic gradient-based optimization of a network-wide learning objective called similarity matching. Our first contribution is generalizing this result to any monotonic activation function by introducing suitable regularizers and constraints to the optimization problem.

Similarity matching is formally defined as follows. Given $T$ inputs, $\mathbf{x}_1, \ldots, \mathbf{x}_T \in \mathbb{R}^K$, and outputs, $\mathbf{r}_1, \ldots, \mathbf{r}_T \in \mathbb{R}^N$, similarity matching learns a representation where pairwise input dot products, or similarities, are preserved subject to regularization and lower and upper bounds on outputs:

$$\min_{\mathbf{r}_1,\ldots,\mathbf{r}_T} \frac{1}{2T^2} \sum_{t=1}^{T} \sum_{t'=1}^{T} \left(\mathbf{x}_t \cdot \mathbf{x}_{t'} - \mathbf{r}_t \cdot \mathbf{r}_{t'}\right)^2 + \frac{2}{T} \sum_{t=1}^{T} \|\mathbf{F}(\mathbf{r}_t)\|_1,$$
$$\text{s.t.} \quad a \leq \mathbf{r}_t \leq b, \qquad t = 1, \ldots, T. \tag{3}$$

Here, the bounds and the regularization function act elementwise.

To see how the Hebbian/Anti-Hebbian network relates to (3), following the method of [14], we expand the square in (3) and introduce new auxiliary variables $\mathbf{W} \in \mathbb{R}^{N \times K}$ and $\mathbf{L} \in \mathbb{R}^{N \times N}$ (which will be related to the corresponding variables in (1) and (2) shortly) using the identities

$$-\frac{1}{T^2} \sum_t \sum_{t'} \mathbf{x}_t^\top \mathbf{x}_{t'} \mathbf{r}_t^\top \mathbf{r}_{t'} = \min_{\mathbf{W} \in \mathbb{R}^{N \times K}} -\frac{2}{T} \sum_t \mathbf{x}_t^\top \mathbf{W}^\top \mathbf{r}_t + \operatorname{Tr} \mathbf{W}^\top \mathbf{W}, \tag{4}$$

$$\frac{1}{2T^2} \sum_t \sum_{t'} \left( \mathbf{r}_t^\top \mathbf{r}_{t'} \right)^2 = \max_{\mathbf{L} \in \mathbb{R}^{N \times N}} \frac{1}{T} \sum_t \mathbf{r}_t^\top \mathbf{L} \mathbf{r}_t - \frac{1}{2} \operatorname{Tr} \mathbf{L}^\top \mathbf{L}. \tag{5}$$

The first line arises from the cross-term in (3), and aligns the similarities in the input to the output. The second line creates diversity in the representation. Plugging these into (3) and exchanging the orders of optimization, we arrive at a dual min-max formulation of similarity matching [14]:

$$\min_{\mathbf{W} \in \mathbb{R}^{N \times K}} \max_{\mathbf{L} \in \mathbb{R}^{N \times N}} \frac{1}{T} \sum_{t=1}^{T} l_t(\mathbf{W}, \mathbf{L}, \mathbf{x}_t), \tag{6}$$

where

$$l_t := \operatorname{Tr} \mathbf{W}^\top \mathbf{W} - \frac{1}{2} \operatorname{Tr} \mathbf{L}^\top \mathbf{L} + \min_{\mathbf{r}_t} \left( -2\mathbf{r}_t^\top \mathbf{W} \mathbf{x}_t + \mathbf{r}_t^\top \mathbf{L} \mathbf{r}_t + 2 \|\mathbf{F}(\mathbf{r}_t)\|_1 \right). \tag{7}$$

The Hebbian/anti-Hebbian network, defined by equations (1) and (2), can be interpreted as a stochastic alternating optimization [17] of the new objective (6). The algorithms performs two steps for each input, $\mathbf{x}_t$.

In the first step, the algorithm minimizes $l_t$ with respect to $\mathbf{r}_t$ by running the neural dynamics (1) until convergence. Minimization is achieved because the argument of $\min$ in (7), $E = -2\mathbf{r}_t^\top \mathbf{W} \mathbf{x}_t + \mathbf{r}_t^\top \mathbf{L} \mathbf{r}_t + 2 \|\mathbf{F}(\mathbf{r}_t)\|_1$, decreases with the neural dynamics (1), if, within the bounds on the output, the regularizer is related to the neural activation function as:

$$F'(r) = u - r, \quad \text{where } r = f(u). \tag{8}$$

The lower and upper bounds $a$ and $b$ are the infimum and supremum of the range of $f$ respectively. We prove a more general version of this result in Proposition 1 in Appendix A. See [24] for other possible neural dynamical systems for $l_t$ minimization.

The following are some examples of the relation (8). The capped rectified linear activation function $f(u) = \min(\max(u - \lambda, 0), b)$ corresponds to a regularization $F(r) = \lambda r + \text{constant}$ with optimization lower and upper bounds $a = 0$ and $b$. When $\lambda = 0$, $a = 0$, $b = \infty$, we recover nonnegative similarity matching [25]. When $F(r) = \text{constant}$, $a = -\infty$ and $b = \infty$, $f(u) = u$ and we recover the principal subspace network of [14]. For other examples of regularizers see [26].

In the second step of the algorithm, synaptic weights are updated by gradient descent-ascent. Given the optimal network output, $\mathbf{r}_t^*$, $\mathbf{W}$ and $\mathbf{L}$ dependent terms in $l_t$ can be written as a distributed summation of local objectives over synapses [14]:

$$\sum_{i=1}^{N} \sum_{j=1}^{K} \left( -2W_{ij} r_{t,i}^* x_{t,j} + W_{ij}^2 \right) + \sum_{i=1}^{N} \sum_{j=1}^{N} \left( L_{ij} r_{t,i}^* r_{t,j}^* - \frac{1}{2} L_{ij}^2 \right). \tag{9}$$

This form explicitly shows how the credit assignment problem is solved in an elegant way. In (9) each term in the summation depends on only local variables to that synapse, gradient descent on $\mathbf{W}$ and gradient ascent in $\mathbf{L}$ results in the local updates given in (2).

## 3 Structured similarity matching

The derivation given in the previous section suggests a generalization of similarity matching in a way that the corresponding Hebbian/anti-Hebbian network has structured connectivity. A close look at equations (9) and (4) reveals that if we modify the left hand side of (4), the input-output similarity alignment term, to

$$-\frac{1}{T^2} \sum_{i=1}^{K} \sum_{j=1}^{N} \sum_{t=1}^{T} \sum_{t'=1}^{T} x_{t,i} x_{t',i} r_{t,j} r_{t',j} c_{ij}^W, \tag{10}$$

where $c_{ij}^W \geq 0$ are constants that set the structure, one can still go through the argument in the previous section and arrive at a modified version (9) where the global objective is still factorized into local objectives. We will do that explicitly in Section 5. A similar modification can be done for the left hand side of (5) by introducing $c_{ij}^L \geq 0$, to arrive at the full structured similarity matching (SSM) cost function

$$\min_{\substack{a \leq \mathbf{r}_t \leq b, \\ t=1,\dots,T}} \frac{1}{T^2} \sum_{t=1}^T \sum_{t'=1}^T \left( -\sum_{i,j} x_{t,i} x_{t',i} r_{t,j} r_{t',j} c_{ij}^W + \frac{1}{2} \sum_{i,j} r_{t,i} r_{t',i} r_{t,j} r_{t',j} c_{ij}^L \right) + \frac{2}{T} \sum_{t=1}^T \|\mathbf{F}(\mathbf{r}_t)\|_1, \tag{11}$$

where we dropped terms that depend only on the input.

Through the choice of $c_{ij}^W$ and $c_{ij}^L$, one can design many topologies for the input-output and output-output interactions. A simple way to choose structure constants is $c_{ij}^W \in \{0,1\}$ and $c_{ij}^L \in \{0,1\}$. Setting $c_{ij}^W = 0$ will remove any direct interaction between the $i^{\text{th}}$ input and $j^{\text{th}}$ output channels, and $c_{ij}^L = 0$ will do the same thing for the corresponding outputs. One can anticipate that such structured similarity matching can be learned by a Hebbian/anti-Hebbian network with corresponding connections removed. We will show this explicitly later in Section 5. Other choices of structure constants assign different weights to particular input-output and output-output interactions. A useful architecture for image processing is the locally connected structure, shown in Figure 1A. It is interesting to note that structured lateral inhibition was also used in [18] for structured sparse coding.

## 4  Structured and deep similarity matching

Next, we generalize structured similarity matching to multi-layer processing. To illustrate the main idea, we first focus on generalizing the original similarity matching objective to multiple layers, and bring in the structure constants later.

We think of a series of similarity matching operations, each applied to the output of the previous layer. For notational convenience, we set $\mathbf{r}_t^{(0)} := \mathbf{x}_t$ and $N^{(0)} := K$, and define deep similarity matching with $P$ layers as:

$$\min_{\substack{a \leq \mathbf{r}_t^{(p)} \leq b, \\ t=1,\dots,T, \\ p=1,\dots,P}} \sum_{p=1}^P \frac{\gamma^{p-P}}{2T^2} \sum_{t=1}^T \sum_{t'=1}^T \left( \mathbf{r}_t^{(p-1)} \cdot \mathbf{r}_{t'}^{(p-1)} - \mathbf{r}_t^{(p)} \cdot \mathbf{r}_{t'}^{(p)} \right)^2 + \sum_{p=1}^P \frac{2\gamma^{p-P}}{T} \sum_{t=1}^T \left\| \mathbf{F}\left( \mathbf{r}_t^{(p)} \right) \right\|_1, \tag{12}$$

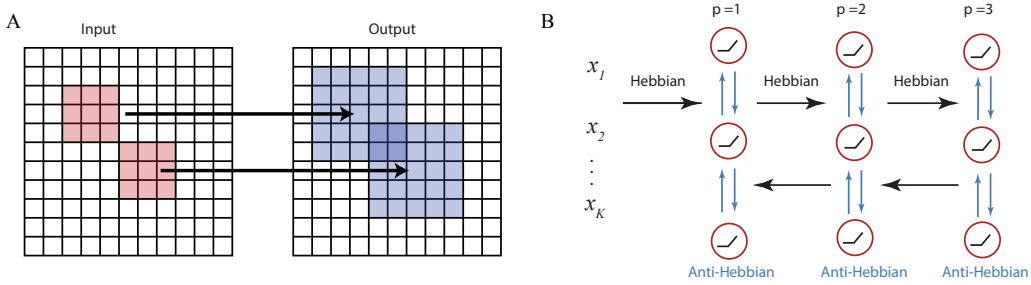

Figure 1: A) Locally connected similarity matching and structured Hebbian/anti-Hebbian network. Choosing the constants $\mathbf{c}^W$ and $\mathbf{c}^L$ suitably, one can introduce structured interactions between inputs and outputs. In this example, we assume inputs $\mathbf{x}$ and outputs $\mathbf{r}$ both live on a 2-dimensional grid. Each output neuron takes input from a small portion of the input grid (red shades) and receives lateral interactions from a subset of the output (blue shades). The corresponding structured Hebbian/anti-Hebbian neural network has the same architecture with connectivity defined by the interactions. Feedforward connections are learned with Hebbian learning rules and lateral connections with anti-Hebbian rules. B) Deep similarity matching and deep Hebbian/anti-Hebbian network. Arrows illustrate synaptic connections. One can introduce structure for all components of the connectivity.

where $\gamma \geq 0$ is a parameter and $\mathbf{r}_t^{(p)} \in \mathbb{R}^{N^{(p)}}$. The $\gamma = 0$ limit corresponds to each layer acting independently on the output of the previous layer. The more interesting case is that of finite $\gamma$, which allows influence of later layers on previous layers. Small $\gamma$ emphasizes costs of earlier layers. In spirit, this construction is similar to deep sparse coding with feedback [20, 21].

One can intuit the kind of network that will optimize the deep similarity matching cost, which we will present a full derivation of in the next section. This will be a multi-layer network with Hebbian learning across layers and anti-Hebbian learning within layers, Figure 1B. An interesting consequence of the finite $\gamma$ coupling between layers will be the existence of feedback connections. When $\gamma = 0$, we obtain a network without any feedback. Previously, Bahroun *et al.* [27] implemented a two-layered similarity matching network without any feedback. This network used biologically-implausible weight sharing by tiling the image plane with identical all-to-all connected networks, and thus is different from our approach.

Different layers in (12) have different representations due to regularization terms and possible changes in dimensionality. To make the framework stronger and allow better hierarchical feature extraction, we introduce nonnegative structure constants $c_{ij}^{W,(p)}$ and $c_{ij}^{L,(p)}$ to each layer and arrive at the structured and deep similarity matching cost function:

$$
\begin{aligned}
\min_{\substack{a \leq \mathbf{r}_t^{(p)} \leq b \\ t=1,\dots,T, \\ p=1,\dots,P}} \sum_{p=1}^{P} \frac{\gamma^{p-P}}{T^2} \sum_{t=1}^{T} \sum_{t'=1}^{T} \Bigg( &- \sum_{i=1}^{N^{(p-1)}} \sum_{j=1}^{N^{(p)}} r_{t,i}^{(p-1)} r_{t',i}^{(p-1)} r_{t,j}^{(p)} r_{t',j}^{(p)} c_{ij}^{W,(p)} \\
&+ \frac{(1+\gamma(1-\delta_{pP}))}{2} \sum_{i=1}^{N^{(p)}} \sum_{j=1}^{N^{(p)}} r_{t,i}^{(p)} r_{t',i}^{(p)} r_{t,j}^{(p)} r_{t',j}^{(p)} c_{ij}^{L,(p)} \Bigg) + \sum_{p=1}^{P} \frac{2\gamma^{p-P}}{T} \sum_{t=1}^{T} \left\| \mathbf{F}\left(\mathbf{r}_t^{(p)}\right) \right\|_1 , \quad (13)
\end{aligned}
$$

where $\delta_{pP}$ is the Kronecker delta. For images, neurobiology suggests choosing the structure constants so that the sizes of receptive fields increase across layers [28].

## 5 Structured and deep similarity matching via structured and deep Hebbian/Anti-Hebbian neural networks

Next, we derive the network that minimizes the structured and deep similarity matching cost (13). We show how credit assignment in this network is solved by explicitly factorizing the dual of the network-wide cost (13) to local synaptic objectives.

Our derivation uses the methods reviewed in Section 2. For each layer, we introduce dual variables $W_{ij}^{(p)}$ and $L_{ij}^{(p)}$ for interactions with positive structure constants, define variables

$$
\bar{W}_{ij}^{(p)} = \begin{cases} W_{ij}^{(p)}, & c_{ij}^{W,(p)} \neq 0 \\ 0, & c_{ij}^{W,(p)} = 0 \end{cases}, \qquad \bar{L}_{ij}^{(p)} = \begin{cases} L_{ij}^{(p)}, & c_{ij}^{L,(p)} \neq 0 \\ 0, & c_{ij}^{L,(p)} = 0 \end{cases}, \qquad (14)
$$

for notational convenience, and rewrite (13) as

$$
\min_{\bar{\mathbf{W}}^{(1)},\dots,\bar{\mathbf{W}}^{(p)}} \max_{\bar{\mathbf{L}}^{(1)},\dots,\bar{\mathbf{L}}^{(p)}} \frac{1}{T} \sum_{t=1}^{T} l_t \left( \bar{\mathbf{W}}^{(1)},\dots,\bar{\mathbf{W}}^{(p)}, \bar{\mathbf{L}}^{(1)},\dots,\bar{\mathbf{L}}^{(p)}, \mathbf{r}_t^{(0)} \right), \qquad (15)
$$

where

$$
\begin{aligned}
l_t := &\sum_{p=1}^{P} \sum_{\substack{i,j \\ c_{ij}^{W,(p)} \neq 0}} \frac{\gamma^{p-P}}{c_{ij}^{W,(p)}} W_{ij}^{(p)2} - \sum_{p=1}^{P} \sum_{\substack{i,j \\ c_{ij}^{L,(p)} \neq 0}} \frac{\gamma^{p-P}}{2\left(1+\gamma(1-\delta_{pP})\right) c_{ij}^{L,(p)}} L_{ij}^{(p)2} \\
&+ \min_{\substack{a \leq \mathbf{r}_t^{(p)} \leq b \\ p=1,\dots,P}} \sum_{p=1}^{P} \gamma^{p-P} \left( -2\mathbf{r}_t^{(p)\top} \bar{\mathbf{W}}^{(p)} \mathbf{r}_t^{(p-1)} + \mathbf{r}_t^{(p)\top} \bar{\mathbf{L}}^{(p)} \mathbf{r}_t^{(p)} + 2\left\| \mathbf{F}\left(\mathbf{r}_t^{(p)}\right) \right\|_1 \right), \quad (16)
\end{aligned}
$$

This new optimization problem can be solved in a stochastic manner, by taking gradients of $l_t$ with respect to $W_{ij}^{(p)}$ and $L_{ij}^{(p)}$, for optimal values of $\mathbf{r}_t^{(p)}$. This procedure is akin to the alternating optimization of sparse coding [17, 29]. We will describe each of these alternating steps separately.

## 5.1 Neural dynamics

Proposition 1 given in Appendix A shows that the minimization of the second line of (16) can be performed by running the following neural network dynamics until convergence to a fixed point,

$$\tau \frac{d\mathbf{u}^{(p)}}{ds} = -\mathbf{u}^{(p)} + \bar{\mathbf{W}}^{(p)}\mathbf{r}^{(p-1)} - \left(\bar{\mathbf{L}}^{(p)} - \mathbf{I}\right)\mathbf{r}^{(p)} + (1 - \delta_{pP})\gamma\bar{\mathbf{W}}^{(p+1)\top}\mathbf{r}^{(p+1)},$$

$$\mathbf{r}^{(p)} = \mathbf{f}(\mathbf{u}^{(p)}), \quad p = 1, \dots, P. \tag{17}$$

where we dropped the $t$ subscript for notational clarity and set $\mathbf{r}^{(0)} = \mathbf{x}_t$. As promised, the $\gamma$ parameter sets the strength of feedback connections in the network. When $\gamma = 0$, information in the network flows bottom up only. In practice waiting until convergence may not be necessary [30].

## 5.2 Gradient-based learning and local learning rules

With the optimal values of the neural dynamics, the network-wide objective factorizes into local synaptic objectives, providing an explicit solution to the credit assignment problem:

$$l_t = \sum_{p=1}^{P}\sum_{\substack{i,j \\ c_{ij}^{W,(p)}\neq 0}}\left(-2W_{ij}^{(p)}r_j^{(p)^*}r_i^{(p-1)^*} + \frac{\gamma^{p-P}}{c_{ij}^{W,(p)}}W_{ij}^{(p)2}\right)$$

$$-\sum_{p=1}^{P}\sum_{\substack{i,j \\ c_{ij}^{L,(p)}\neq 0}}\left(-L_{ij}^{(p)}r_j^{(p)^*}r_i^{(p)^*} + \frac{\gamma^{p-P}}{2\left(1 + \gamma(1 - \delta_{pP})\right)c_{ij}^{L,(p)}}L_{ij}^{(p)2}\right). \tag{18}$$

Local learning rules are derived from the above equation by taking derivatives:

$$\Delta W_{ij}^{(p)} = \eta\gamma^{p-P}\left(r_j^{(p)^*}r_i^{(p-1)^*} - \frac{W_{ij}^{(p)}}{c_{ij}^{W,(p)}}\right),$$

$$\Delta L_{ij}^{(p)} = \frac{\eta}{2}\gamma^{p-P}\left(r_j^{(p)^*}r_i^{(p)^*} - \frac{L_{ij}^{(p)}}{\left(1 + \gamma(1 - \delta_{pP})\right)c_{ij}^{L,(p)}}\right). \tag{19}$$

These rules are Hebbian between layers and anti-Hebbian within layers, Figure 1. One can absorb the $\gamma$ factors into the learning rates and choose different rates for different layers for better performance.

Equations (17) and (19) define the structured and deep Hebbian/anti-Hebbian neural network, Figure 1B. It operates by running the multi-layered dynamics (17) for each input, and performing the updates (19) before seeing the next input.

# 6 Simulations

Next, we illustrate the performance of the structured and deep similarity matching networks in various datasets.

## 6.1 Illustrative example

We start by introducing a toy example that illustrates the operational principles of the structured and deep Hebbian/anti-Hebbian neural network. We trained a two-layer network with the following architecture: the first layer is composed of two separate networks of 10 neurons each, while the second layer is composed of a single network of 10 neurons. The second layer is connected to both first layer networks with feedforward and feedback ($\gamma = 0.8$) connections, Figure 2. Inputs to the network are clustered into two groups, and are drawn randomly from 100-dimensional Gaussian distributions. Representational similarity of these patterns are shown in Figure 2. The Gaussian distributions were chosen separately for each cluster and first layer network. Neural activation functions were $f(a) = \max(\min(a, 1), 0)$. We used a regularized version of the similarity matching cost [31] to enforce pattern decorrelation in the first and second layers. This regularization does not

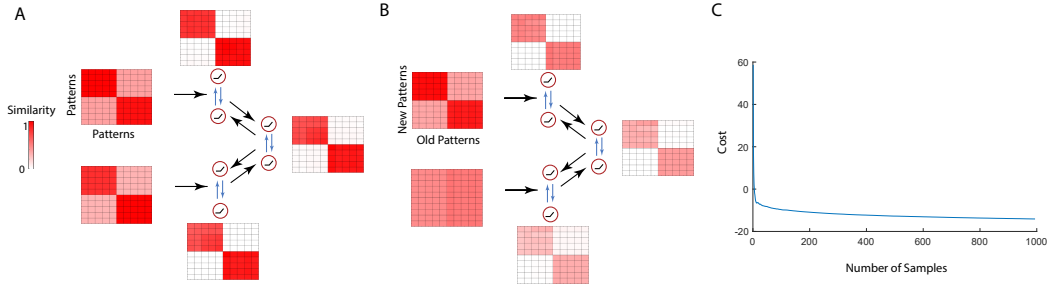

Figure 2: A two-layer Hebbian/anti-Hebbian network with feedback. For each subnetwork, representational similarity matrices are shown for 10 example patterns, 5 from each of the two clusters. Similarities are calculated by taking pairwise dot products of patterns and normalizing the largest dot product to 1. A) Network simulated with patterns from a set generated from the same distribution as the training set. B) Network simulated with patterns to the bottom first layer generated from a different distribution. C) Structured and deep similarity matching cost decreases over training.

.

change the biological plausibility of our networks, just adds homeostatic plasticity rules [31, 32]. During training, the structured and deep similarity matching cost consistently decreased 2C. The networks learned decorrelated representations in the first and second layers, 2A.

Next, we performed a perturbation that elucidates the role of feedback. We kept the input distribution to one of the first layer networks (top in 2B) as is, but changed the inputs to the other first layer network (bottom 2B). The new patterns were nearly equally similar to the original clusters of patterns (bottom 2B). Even though the cluster identity of these new patterns were ambiguous, the bottom network clustered them to the first or second cluster, depending on the identity of inputs to the top network. This decision was mediated by the feedback connections from the second layer. Therefore, while the anti-Hebbian connections within a layer were performing competitive learning and pattern separation, the Hebbian connections between layers were creating a predictive, pattern completing pathway between the hierarchical representations across layers.

## 6.2  Faces dataset

We trained a 3-layer, locally connected Hebbian/anti-Hebbian neural network with examples from the "labeled faces in the wild" dataset [33], Figure 3. Images in this dataset have dimensions $64^2$. We organized the neurons into square grids in each layer, with strides 2, 4 and 8 respectively in first, second and third layer. Thus, there were 1024, 256 and 64 neurons in respective layers. A neuron was connected to a neuron in the previous layer if the Euclidean distance between its grid location and the previous layer neuron's grid location was less than or equal to 8 for the first layer, 12 for the second layer and 24 for the third layer. Lateral connections were again based on Euclidean distances with the same parameters. We trained with different $\gamma$ values, shown are features for $\gamma = 0.01$. Figure 3 shows the learned features. Neural activation functions were $f(a) = \max(\min(a, 1), 0)$. We see that the network learns diverse localized features in the first layer, and combines them in the second and third layers to larger scale features.

## 6.3  Classifying hand-written digits

We next tested if the features learned by our networks are useful for classification tasks. We trained a single-layer structured similarity matching network on the MNIST data set, with each image preprocessed by mean subtraction. We used the locally-connected structure shown in Figure 1. Network had a stride 2 and each neuron received input from a patch of radius $r_o = 4$. Neurons belonging to the same site had inhibitory recurrent connections. We used hyperbolic tangent activation function (tanh(x)). Classification was done using scikitlearn library's LinearSVC with default parameters. Table 1 shows classification error as a function of number of neurons per site (NPS). When compared to other networks with biologically-plausible training (1.46% in [34]), our network achieves on-par performance on this dataset.

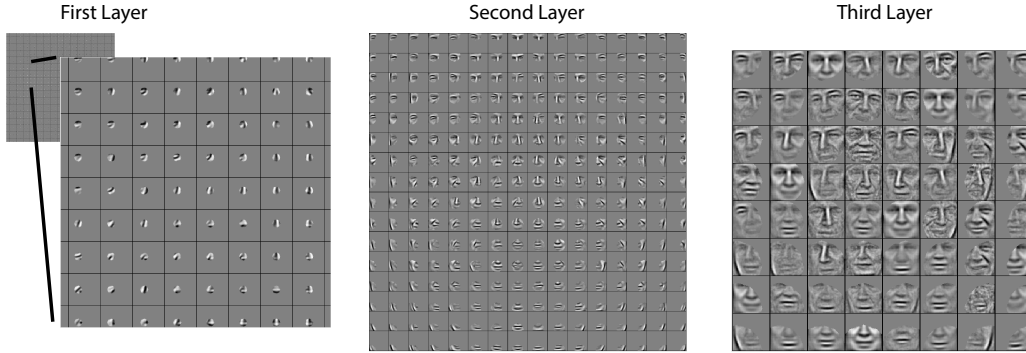

First Layer          Second Layer          Third Layer

Figure 3: Features learned by a 3-layer, locally connected Hebbian/anti-Hebbian neural network on the labeled faces in the wild dataset [33]. Features are calculated by reverse correlation on the dataset, and masking these features to keep only the portions of the dataset which elicits a response in the neuron. We zoom to a selected subset of features for the first layer on the left.

| NPS | 4 | 8 | 16 | 32 | 64 | 100 |
|---|---|---|---|---|---|---|
| Classification error (%) | 3.87 | 2.41 | 1.73 | 1.60 | 1.47 | 1.40 |

Table 1: Classification on MNIST data set: we show how the test error decreases as the number of neurons per site (NPS) increases.

# 7  Discussion and conclusion

We introduced a new class of unsupervised learning cost functions, structured and deep similarity matching, and showed how they can efficiently be minimized via a new class of neural networks: structured and deep Hebbian/anti-Hebbian networks. These networks generalize Földiak's single layer, all-to-all connected network [22].

Even though we introduced depth separately from structure within a layer, they are actually related. The structured and deep cost function in (13) can be obtained from the structured cost function (11) by allowing structure constants to be negative, and choosing them and regularizers suitably. Our framework can be used to introduce other architectures, e.g. ones including skip connections.

The credit assignment problem in our networks is solved in an efficient manner. Through a duality transform [14], we showed how the dual min-max objective is factorized into distributed objectives over synapses, that depend only on variables local to that synapse. Therefore, each synapse can be updated by biologically-plausible local learning rules and yet the global objective can be optimized in a gradient-based manner.

There are two possible "weight transport problems" [35] in our networks: 1) feedback connections are transposes of feedforward connections, and 2) lateral connections are symmetric. A straightforward and biologically-plausible solution to these problems exist: symmetric weights can be learned asymptotically by the local learning rules in (19), even when the weights are initialized differently. A similar solution was proposed in [36] to the weight transport problem in the backpropagation algorithm. Other solutions, including random feedback weights [5], may be possible.

**Acknowledgments**

We thank Alper Erdogan and Blake Bordelon for discussions. This work was supported by a gift from the Intel Corporation.

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
