[Supplementary Material · NIPS_2019_supplement.pdf]

# Supplementary Material

## A  Proof of Proposition 1

**Proposition 1.** *Assume $f$ is monotonically increasing and bounded. Define*

$$E := \sum_{p=1}^{P} \gamma^{p-P} \left( -2\mathbf{r}^{(p)\top}\bar{\mathbf{W}}^{(p)}\mathbf{r}^{(p-1)} + \mathbf{r}^{(p)\top}\bar{\mathbf{L}}^{(p)}\mathbf{r}^{(p)} + 2\left\|\mathbf{F}\left(\mathbf{r}^{(p)}\right)\right\|_1 \right). \tag{20}$$

*Then, under the dynamics* (17)*, $E$ is bounded from below and nonincreasing:*

$$\frac{dE}{ds} \leq 0 \tag{21}$$

*If $f'(u) > 0$, then*

$$\frac{dE}{ds} = 0 \tag{22}$$

*if and only if at the fixed points.*

*Proof.*  By chain rule

$$\frac{dE}{ds} = \sum_{p=1}^{P} \frac{\partial E}{\partial \mathbf{r}^{(p)}} \frac{d\mathbf{r}^{(p)}}{ds}$$

$$= -2\sum_{p=1}^{P} \gamma^{p-P} \left[\bar{\mathbf{W}}^{(p)}\mathbf{r}^{(p-1)} - \bar{\mathbf{L}}^{(p)}\mathbf{r}^{(p)} + \gamma(1-\delta_{p,P})\bar{\mathbf{W}}^{(p+1)\top}\mathbf{r}^{(p+1)} - \mathbf{F}'\left(\mathbf{r}^{(p)}\right)\right] \cdot \frac{d\mathbf{r}^{(p)}}{ds}$$

$$= -\frac{2}{\tau}\sum_{p=1}^{P} \gamma^{p-P} \frac{d\mathbf{u}^{(p)}}{ds} \cdot \frac{d\mathbf{r}^{(p)}}{ds} = -2\sum_{p=1}^{P} \gamma^{p-P} \sum_{i=1}^{N^{(p)}} \left(\frac{du_i^{(p)}}{ds}\right)^2 f'(u_i^p)$$

$$\leq 0, \tag{23}$$

where the last inequality holds because $f$ is monotonically increasing. $E$ is nonincreasing and bounded from below because $f$ are bounded. If $f'(u) > 0$, then $E$ is stationary if and only if at the fixed points and therefore $E$ is a Lyapunov function.

Similar proofs were given in e.g. [37, 3, 26]. $\qquad\square$

**Note 1.**  *When $P = 1$, for the all-to-all connected Hebbian/anti-Hebbian network of Section 2, the activation function does not need to be bounded for $E$ to be lower bounded, if $\mathbf{L}$ is initialized positive definite. The learning rules* (2) *preserve positive definiteness of $\mathbf{L}$.*

The energy function (20) and the corresponding dynamics (17) resembles the ones used in Xie and Seung's Contrastive Hebbian Learning (CHL) [3]. We want to take the opportunity to discuss the differences between the two approaches. Most importantly, CHL optimizes an error defined at the output, and therefore has to (back)propagate the error by feedback connections. In deep similarity matching error is defined as a function of all neurons at all layers, through duality can be reduced to a local error for each synapse. In this sense, there is nothing special about feedback connections. Even without the feedback ($\gamma = 0$ limit), each layer performs gradient-based learning. Some other differences are: 1) CHL performs approximate gradient-descent. Our network performs exact gradient descent-ascent. 2) CHL does not have lateral connections within a layer, our network does. 3) CHL has two (clamped and unclamped) phases for neural dynamics, our network has only one.