[Reviews · NeurIPS 2019]

Reviewer 1



EDIT: I have read the author response and appreciate the effort made by the authors' to address review suggestions. Unfortunately, the empirical results given in the author response appear to detract from, rather than add to, the significance of the work. Deep networks produce less useful features than shallow networks. Since the similarity-matching-for-shallow-networks method was already present in the literature, it is unclear what benefit the method being presented authors. The authors provided no comparison between their structured network approach and an unstructured network, so it is not possible to tell if their other contribution provides any empirical benefit. Without such a contribution, I find it difficult to recommend this paper in its current form for publication, particularly in light of the point mentioned in my initial review that the paper's main conceptual insights are drawn from prior work. Further experimentation or refinement of the method may address this issue, and I would love to see an updated paper that does so. ORIGINAL REVIEW: This paper generalizes recent work that connects similarity matching to particular Hebbian / Anti-Hebbian dense and shallow network architectures, to the structured and deep network case. The mathematical derivation is clear, particularly in the context of the papers it builds on. However, I have concerns about the novelty and significance of the result. 1. The feedback architecture is essentially the same as that used in contrastive Hebbian learning, a model not mentioned explicitly in the paper though cited implicity (citation 20, Xie & Seung). Moreover, Xie & Seung proved that contrastive Hebbian learning can approximate (or in a certain regime be equivalent to) backpropagation. Thus this paper appears to, in effect, stitch together the insight of the earlier literature on relating similarity matching objectives to Hebbian / anti-Hebbian learning rules and the insight of older literature on the ability of contrastive Hebbian learning to use feedback and local learning rules to assign credit in deep neural networks. Moreover, Bahroun et al. arrived at a similar architecture excluding the feedback component. To me, this scale of theoretical contribution warrants publication only if it leads to meaningful empirical advances, which (see below) I don't find to be demonstrated in the paper in its current form. 2. More explanation is warranted for the value of similarity matching as an objective for deep networks. After all, the goal of many deep learning problems is to radically warp the similarity metric between data points! Similarity matching in the linear case has a nice interpretation in terms of PCA, but the authors do not give a similar interpretation to the present model (e.g. a relationship to autoencoders). 3. The empirical results are uncompelling. First, the role and value of feedback, which is the main contribution of this paper beyond Bahroun et al., is not demonstrated empirically. Second, the features learned by deep networks are not compared to those learned by a comparable shallow model. Thirdly, no quantitative evidence of the quality of the learned features (e.g. their utility in a downstream supervised learning task) is given. 4. This paper does not claim to aim for state-of-the-art empirical results, but rather to introduce a biologically plausible approach to unsupervised learning. As such, the biological plausibility of the model is very important. The proposed model involves an obvious weight transport issue, which the authors acknowledge and propose a workaround for (namely, that Hebbian learning of both forward and backward weights will tend to cause them to become transposes of one another). While an interesting argument, I am not entirely convinced that it is robust to biological noise, other sources of learning, etc.

Reviewer 2



——Rebuttal Response—— It looks like my score is a bit of an outlier here. I took some time to read one more time the paper, all the discussions, and the author’s rebuttal. While I hear the objections, my position remains unchanged. This is a very interesting paper and I think that it should be accepted. This paper is indeed a first look at the problem, rather than a fully conclusive and executed research program, but I think it contains an interesting idea. Compared to the previous work on similarity matching, this is the first one that extends similarity matching cost function to networks with several layers and structured connectivity. In my view, this is a huge step. It is also nice that the authors provide a reasonably “biological” implementation of their optimization, yes, with a little bit of weight transport, but nothing is perfectly biological anyway. Regarding table 1 of the rebuttal. I would encourage the authors to invest enough time to experiment with the hyperparameters and clearly state in the final version where they stand. If it turns out that deep architectures learn representations leading to a better accuracy, this is an interesting statement. If it turns out that shallow architectures lead to a better accuracy, this is an equally interesting and important statement. Since the proposed algorithm is different from the traditional backpropagation learning both results are possible. I agree with reviewer 3 that the paper is hard to follow. I would encourage the authors to invest time in making the presentation more clear, as they promised in the rebuttal. ———————— There is a large body of work devoted to approximating backpropagation algorithm by local learning rules with varying degree of biological plausibility. On the other hand, the amount of work dedicated to investigating various biologically-plausible learning rules and their computational significance for modern machine learning problems is rather small. This paper tackles the second problem and demonstrates that a network with structured connectivity and a good degree of biological plausibility is capable of learning latent representations. This is a significant statement. For this reason I argue in favor of accepting this paper. Specifically the authors study similarity-matching objective function, and show how it can be optimized in deep networks with structured connectivity. To the best of my knowledge these results are new. While the general idea and results are clear, I have several technical questions throughout the paper: 1. There seem to be a misprint in Eq (2). Is y_i the same variable as r_i? 2. I do not understand the “Illustrative example” in section 6.1 How were the inputs constructed? What does it mean that they were “clustered into two groups and drawn from a gaussian distribution”? There are many ways how one can do this. Is it possible to show examples of the inputs? Without this, it’s hard to tell how non-trivial these results are. In general, this section would benefit from more detailed explanations, I think. 3. Figure 3 is impressive, but the description of how it is constructed is insufficient, in my opinion. I would appreciate a clear explanation of the sentence “Features are calculated by reverse correlation on the dataset, and masking these features to keep only the portions of the dataset which elicits a response in the neuron”. Maybe in supplementary materials? 4. What is the interpretation of the features in the first layer? Are they PCA components of the inputs or are they something else? 5. I understand that learning rules (2) are needed for the optimization of the similarity-matching (3). Out of curiosity, would something like figure 3 emerge if in equation (1) the weights L were fixed and all equal to a positive constant (global lateral inhibition), or is it crucial that the weights L are learned? In a recent paper https://www.pnas.org/content/116/16/7723, for example, a network with constant L was shown to lead to high quality latent representations. Are the representations reported in this submission in some sense “better” than the ones reported in the above paper (since the weights L are learned), or is learning the weights L simply necessary to make a connection with the similarity matching objective function?

Reviewer 3



*** REBUTTAL RESPONSE *** Thank you for the sincere rebuttal. I've changed my review to marginally above accept. I look forward to revisions for clarity and to the expanded empirical evaluations as mentioned in the rebuttal. *** ORIGINAL REVIEW *** As observed in previous studies certain similarity matching objectives leads to a biologically plausible (local) hebbian/anti-hebbian learning algorithm. This paper introduces a "deep" and "structured" similarity matching objective and derive the corresponding local hebbian/anti-hebbian learning algorithm. The objective is structured in the sense that constraints are introduced on which input and output neurons pairs should contribute to the matching objective. In particular for image inputs local spatial structure is imposed, akin to convolutional filters. The structured objective is a generalization and becomes the unstructured objective in the limit of no constraints. The similarity matching objective is deep in the sense that its the sum of similarity measures over multiple layers of a multi-layer neural network. The paper performs experiments on a toy dataset and on labeled faces in the wild. The main results are the filters learned by the network, which resemble a clear hierarchy of face parts, although it's not clear how the filters were visualized exactly. w.r.t. originality: In a sense the deep and structured extensions seem to be "just" imported and brought together from previous studies in sparse coding (no small feat to be sure). It is however very interesting to see what this deep, structured hebbian/anti-hebbian learning is actually optimizing for. w.r.t. quality and clarity: I think the paper is very hard to follow and has several quality issues. In my opinion it could be a lot better. The introduction and abstract are fairly well written, and explains the idea pretty well. However in the following sections much of the math is poorly introduced and hard to follow. For instance W and L are not defined in eq. 1 and y_i is not defined in eq. 2. eq. 2 is given as self-evident, but isn't shown until eq. 9. A "leak term" is mentioned on line 77 which is never defined. I couldn't follow the parts on regularization in eq. 3, and 4 where several results are given as self-evident. At this point in the text the link between eq. 1, 2 and 3 have not yet been made, but the regularization discussion mentions these links as self-evident and makes extensive claims. I'd suggest the authors either take the time and space needed to actually introduce and explain the equations, or simply give them as proven, give the relevant reference and explain what they mean. If you read the referenced papers it becomes more and more clear what the author means by their math, but the paper is simply not understandable as a self contained unit in my opinion. After having spent 2 days trying to understand section 2 I gave up, and skimmed the rest of the math, and tried to understand the idea instead. Maybe I'm just not intelligent enough, although I think I'm fairly close to the average reader, and I think I was more patient than the average reader would be. I suggest significant editing of the manuscript, with a focus on how understandable it is to the average reader and what parts of the math are truly significant to the idea. Nitpick: It's "The credit assignment problem is ..." not "Credit assignment problem is ..." The experimental results looks good, but are presented without much analysis. The paper introduces several hyper-parameters, e.g. the gamma parameter which determines feedback (top down) strength, the regularization functions, the structural constraints, etc. but doesn't examine how these affect the results or whether they're important. The results are not compared to any other methods and it's not exactly clear how the experiments were performed. I'd not be able to reproduce the experiments. I want to like this paper, since I think the subject is fascinating, and the direction important, but in the current form I can't recommend it for publication.

[Author Response · NeurIPS 2019]

**Reviewer 1:** We thank the reviewer for a thorough review and kindly request a reevaluation considering the following.

*Relation to Xie and Seung's contrastive Hebbian learning (CHL):* Indeed a missed opportunity! Thank you for bringing this up. We will provide a detailed comparison. However, we think the relation is not as direct as the reviewer suggests. In deep similarity matching, error is defined as a function of all neurons at all layers, and through duality can be reduced to a local error for each synapse. In this sense, there is nothing special about feedback connections. Even without the feedback ($\gamma = 0$ limit), each layer is doing gradient-based learning. This is in contrast to CHL, which optimizes an error defined at the output, and therefore has to (back)propagate it by feedback connections. Some other differences are: 1) CHL performs approximate gradient-descent. Our network performs exact gradient descent-ascent. 2) Our network has lateral connectivity, CHL does not. 3) CHL has clamped and unclamped phases, our network does not.

*Similarity matching as a deep learning objective function:* This is a very fair criticism and we should have been very clear about it in our paper. The name similarity matching suggests that pairwise similarities are preserved across layers, but in reality, because of the structured connectivity and regularization, the similarity structure changes in a very nonlinear way. For example, previous work showed that (unstructured) nonnegative similarity matching can be interpreted as nonnegative ICA or manifold learning. By introducing structure and depth, surely the network's power to warp similarity metrics will increase. An analytical understanding of how exactly such warping will happen is hard, and should be the topic of later papers. Our simulations were chosen to give some intuition: one illustrates how feedback can provide associative links, and the other one shows hierarchical feature extraction. We thank the reviewer for the autoencoder connection, which we will make.

*Comparison to Bahroun et. al.* While we appreciate Bahroun *et. al.*'s important contribution, our paper's scope goes much beyond it. To be technically correct, Bahroun *et. al.*'s network uses (biologically-implausible) weight sharing, and therefore it is basically a repeated

| (NpS, $d_l$) | (4,0) | (4,4) | (16,0) | (32, 0) |
|---|---|---|---|---|
| error (%) | 4.96 | 4.59 | 2.79 | 2.05 |
| $\gamma$ | 0 | 0 (n) | 0.025 | 0.025 (n) |
| error (%) | 8.12 | 9.53 | 7.67 | 8.85 |

Table 1: MNIST classification (test error) by a linear classifier trained of representations learned by a locally connected network. Top: single-layer, bottom: two-layers. Neurons are organized with a stride of 2 and multiple neurons (or features) per site (NpS), got feedforward input from pixels within radius 4, and lateral input from neurons within radius $\leq d_l$. Two-layer simulations had NpS = 2 and $d_l = 4$ for the first layer and $8$ for the second. (n) denotes test set (not training) with occluded 3x3 patches. Increasing NpS and $d_l$ increased the performance of the classifier. Optimal $\gamma$ is $> 0$.

set of (unstructured) similarity matching networks tiling an image. This is different than our locally structured network, which can learn different features for different portions of an image, while possibly having long-range lateral interactions. When citing Bahroun *et. al.* we did not make this difference very clear, and probably caused the reviewer's confusion. We apologize for that. More importantly, we provide global objective functions for a much larger family of structured and deep architectures (with local learning), locally-connected being just one example.

*Empirical results:* All reviewers correctly asked us to asses the quality of our learned features by a classification task. We launched a detailed numerical study, some preliminary results on MNIST is in Table 1. CIFAR-10 will be added.

*Weight transport:* We appreciate the concern about our solution's robustness. We haven't done a detailed robustness analysis (except to initialization), and we will discuss this point. We recently learned about similar ideas in the backpropagation literature, which we will cite (Kevin and Pollack 1994, Akrout *et. al.* 2019).

**Reviewer 2:** We thank the reviewer for the enthusiastic support! In Eq. (2) $y_i$ is the same variable as $r_i$. Sorry!

*Illustrative example:* We will provide details here, thanks! To be fair, all relevant information about the dataset for similarity matching purposes is already in the shown similarity matrices in the figure (input data have dot products of $\sim 0.5$ across clusters, and $\sim 1$ within the cluster), but we should have made our data generation clear and will do so.

*Figure 3:* We used a common technique from neuroscience for visualization: reverse correlation, which is an average of the input images weighted by a neuron's response. We then set to zero the portions of the image that will not elicit a response in the neuron, because of the limited range of connectivity. We will expand on this in an appendix.

Interpretation: Closest would be ICA due to nonlinearity, please also see response to Reviewer 1.

*Classification results:* Please see our response to Reviewer 1 and preliminary results in Table 1.

*Global lateral inhibition:* This is an excellent suggestion! With fixed weights, the network ends up solving a modified similarity matching problem, Eq. (8) with L fixed. It is not clear to us whether one approach is better than the other. We will cite the Krotov, Hopfield reference (which we should have already done) and discuss this point.

**Reviewer 3:** We thank the reviewer for the sincere feedback, which was truly a wake up call. We were embarrassingly reminded that our mathematical notation and terminology that was lucid to us because of our scientific background may not be so for others. We will revise our paper with this in mind, and take into account all your suggestions. In particular, section 2 was meant to be a short review of previous papers augmented with new findings. This strategy does not work. We will expand section 2, and others, to make the paper self-contained, moving technical details to the appendix.

*Hyperparameters and empirical results:* Please see our response to Reviewer 1 and preliminary results in Table 1.

*Originality:* The high level ideas of depth and structured connectivity have existed much before they appeared in the sparse coding literature. Our contributions are 1) implementing these ideas in the similarity matching framework at the cost function level, and 2) providing biologically-plausible networks that can optimize such cost functions.

[Meta-Review · NeurIPS 2019]

This paper presents a local learning rule for similarity matching in multilayer neural networks. Similarity matching means that the pairwise dot products of input vectors are preserved in output vectors. The authors factorize the global similarity matching cost function and use this to develop local cost functions for each synapse. They generalize this to deep structured networks, and derive a learning algorithm with Hebbian training between layers and anti-Hebbian training within layers. The authors demonstrate that these networks learn potentially useful hierarchical features. The reviewers agreed that this paper provided a novel, elegant algorithmic contribution. There were concerns that the authors did not successfully show that the hierarchical representations learned actually provided any utility, but it was decided in discussion that despite this concern, the paper was interesting enough to accept.